# BMP-9 Modulates the Hepatic Responses to LPS

**DOI:** 10.3390/cells9030617

**Published:** 2020-03-04

**Authors:** Haristi Gaitantzi, Julius Karch, Lena Germann, Chen Cai, Vanessa Rausch, Emrullah Birgin, Nuh Rahbari, Tatjana Seitz, Claus Hellerbrand, Courtney König, Hellmut G. Augustin, Carolin Mogler, Carolina de la Torre, Norbert Gretz, Timo Itzel, Andreas Teufel, Matthias P. A. Ebert, Katja Breitkopf-Heinlein

**Affiliations:** 1Department of Medicine II, Medical Faculty Mannheim, Heidelberg University, 68167 Mannheim, Germany; haristi.gaitantzi@medma.uni-heidelberg.de (H.G.); julius.karch@gmx.de (J.K.); lenagermann@gmx.at (L.G.); chen.cai@medma.uni-heidelberg.de (C.C.); timo.itzel@medma.uni-heidelberg.de (T.I.); andreas.teufel@medma.uni-heidelberg.de (A.T.); matthias.ebert@medma.uni-heidelberg.de (M.P.A.E.); 2Center for Alcohol Research, University of Heidelberg and Salem Medical Center, 69120 Heidelberg, Germany; rausch.vanessa@gmail.com; 3Department of Surgery, University Medicine Mannheim, Medical Faculty Mannheim, Heidelberg University, 68167 Mannheim, Germany; emrullah.birgin@umm.de (E.B.); nuh.rahbari@umm.de (N.R.); 4Institute of Biochemistry, Emil-Fischer-Zentrum, Friedrich-Alexander-University Erlangen-Nürnberg, 91054 Erlangen, Germany; tatjana.seitz@fau.de (T.S.); claus.hellerbrand@fau.de (C.H.); 5Division of Vascular Oncology and Metastasis, German Cancer Research Center Heidelberg (DKFZ-ZMBH Alliance), 69120 Heidelberg, Germany; courtney.koenig@hotmail.de (C.K.); h.augustin@dkfz-heidelberg.de (H.G.A.); 6European Center for Angioscience (ECAS), Medical Faculty Mannheim, Heidelberg University, 68167 Mannheim, Germany; 7Institute of Pathology, Technical University of Munich, 80333 Munich, Germany; carolin.mogler@tum.de; 8Medical Faculty Mannheim, Medical Research Center, Heidelberg University, 68167 Mannheim, Germany; carolina.delatorre@medma.uni-heidelberg.de (C.d.l.T.); norbert.gretz@medma.uni-heidelberg.de (N.G.)

**Keywords:** LPS, BMP-9, HSC, LSEC, kupffer cells, liver, capillarization, IL-6, macrophages, myofibroblasts

## Abstract

It was previously shown that Bone Morphogenetic Protein (BMP)-9 is constitutively produced and secreted by hepatic stellate cells (HSC). Upon acute liver damage, BMP-9 expression is transiently down-regulated and blocking BMP-9 under conditions of chronic damage ameliorated liver fibrogenesis in C57BL/6 mice. Thereby, BMP-9 acted as a pro-fibrogenic cytokine in the liver but without directly activating isolated HSC in vitro. Lipopolysaccharide (LPS), an endotoxin derived from the membrane of Gram-negative bacteria in the gut, is known to be essential in the pathogenesis of diverse kinds of liver diseases. The aim of the present project was therefore to investigate how high levels of BMP-9 in the context of LPS signalling might result in enhanced liver damage. For this purpose, we stimulated human liver sinusoidal endothelial cells (LSEC) with LPS and incubated primary human liver myofibroblasts (MF) with the conditioned medium of these cells. We found that LPS led to the secretion of factors from LSEC that upregulate BMP-9 expression in MF. At least one of these BMP-9 enhancing factors was defined to be IL-6. High BMP-9 in turn, especially in combination with LPS stimulation, induced the expression of certain capillarization markers in LSEC and enhanced the LPS-mediated induction of pro-inflammatory cytokines in primary human macrophages. In LSEC, pre-treatment with BMP-9 reduced the LPS-mediated activation of the NfkB pathway, whereas in macrophages, LPS partially inhibited the BMP-9/Smad-1 signaling cascade. In vivo, in mice, BMP-9 led to the enhanced presence of F4/80-positive cells in the liver and it modulated the LPS-mediated regulation of inflammatory mediators. In summary, our data point to BMP-9 being a complex and highly dynamic modulator of hepatic responses to LPS: Initial effects of LPS on LSEC led to the upregulation of BMP-9 in MF but sustained high levels of BMP-9 in turn promote pro-inflammatory reactions of macrophages. Thereby, the spatial and timely fine-tuned presence (or absence) of BMP-9 is needed for efficient wound-healing responses in the liver.

## 1. Introduction

The transforming growth factor (TGF)-β superfamily accounts for a number of different cytokines such as TGFβs, activins, nodal, bone morphogenetic proteins (BMPs), growth and differentiation factors (GDF) and the anti-Müllerian hormone [1]. BMP-9 has been reported to be mainly expressed in liver tissue [2] and to play a key role in osteogenesis, liver fibrosis, angiogenesis and cancer [3,4,5,6]. BMP-9 binds to BMP type I receptors like activin-receptor like kinase (ALK1) or Activin A Receptor Type *1* (ALK2) on the cell membrane and promotes their phosphorylation by BMP type II receptors (BMPRII, activin receptor II (ActRII) A and ActRIIB) to activate canonical downstream signalling pathways via the phosphorylation of Smad1, 5, and 8 (R-Smads). The binding of BMP-9 to ALK1 occurs with a much higher affinity than to ALK2 [7]. Together with Smad4, these R-Smads are translocated into the nucleus to modulate the expression of BMP-9-target genes [1]. BMP-9 can also activate non-canonical pathways (non-Smad pathways) such as the p38 mitogen-activated protein kinase (MAPK) pathway, the c-Jun N-terminal kinase (JNK) pathway or the Akt/phosphatidylinositol-4,5-bisphosphate 3-kinase (PI3K) pathway. However, the exact mechanisms of activation of these pathways remain poorly understood [1].

We have previously shown that high levels of BMP-9, produced by hepatic stellate cells (HSC), promote liver fibrosis and counter-act hepatic regenerative processes [5] and that BMP-9 aggravates steato-hepatitis in mice fed a methionine choline deficiency diet [8]. We could further show that the high affinity BMP-9 receptor, ALK-1, is mainly expressed in liver sinusoidal endothelial cells (LSEC) and Kupffer cells (KC) but less in HSC and hepatocytes [5]. In the present study we aimed at further investigating the molecular mechanisms of hepatic BMP-9 effects. Previously, we did not observe any direct HSC activating properties of BMP-9 in vitro [5], implying that its pro-fibrogenic effects in vivo might be of a rather indirect nature. During hepatic wound healing responses, almost all other liver cell types are involved in regulating HSC quiescence or activation [9]. We therefore tested the hypothesis that BMP-9 mediated HSC activation in vivo might be the result of cross-talk between the different non-parenchymal liver cell types (LSEC, KC, HSC, and other fibroblast subtypes).

Lipopolysaccharide (LPS) is an endotoxin derived from the cell walls of gram-negative bacteria in the gut. Due to changes in the intestinal mucosal permeability LPS levels in the portal and systemic circulation increase upon liver injury [10]. Via signalling to its specific receptor, TLR4, LPS can aggravate inflammation and fibrosis in diverse models of liver damage [11]. Like BMP-9, LPS does not directly induce activation of HSC in vitro, but sensitizes the cells for pro-fibrogenic responses to TGF-β [11].

The direct effects of LPS on HSC as well as the cross-talk between LPS-stimulated KC and HSC has been quite well investigated in the past and it was convincingly shown that LPS mediates the initial steps of hepatic fibrogenesis in many models of liver damage. However, in this context, the potential role of LSEC has been less well investigated and very little is known about the potential functions of BMP-9 in orchestrating hepatic LPS effects. In general, BMP-9 was described to control vascular quiescence [12,13] and it increases monocyte recruitment to TNF-α-treated human aortic endothelial cells (HAECs) [14]. We therefore addressed the question of whether the cross-talk between LSEC and other cell types changes BMP-9 expression, and if BMP-9 in turn modulates the hepatic responses to LPS.

## 2. Materials and Methods

Isolation of mouse liver cells. Hepatocytes, HSC, LSEC, and KC were isolated by in situ collagenase perfusion of healthy mouse livers (C57/Bl6) and were immediately processed for PCR analyses as described previously [5].

Isolation of primary human HSC, MF, KC, HC, and LSEC. Primary human liver cells were isolated from fresh tissue samples obtained from the non-malignant liver parts surrounding a colorectal cancer (CRC) liver metastasis. The livers/patients were tested negative for hepatic viral infection. Informed consent was obtained from all patients, and tissue procurement was approved by the local Medical Ethics Committees (2012-293N-MA). Ca. 20 g of tissue was perfused with pre-warmed Perfusion Solution I (Perfusion Solution 10x: (41.5 g NaCl, 2.5 g KCL, 12 g HEPES dissolved in 500 mL ddH_2_O, pH adjusted to 7.5; 950 mg EGTA, 816 mg N-Acetyl-L-cysteine, filled to 1 L, pH adjusted to 7.5) for 30 min followed by perfusion with Collagenase Solution (= Perfusion Solution II: [5.85 g NaCl, 0.75 g KCl, 36 g HEPES, 7.5 g BSA; filled to 1300 mL ddH_2_O, 1.05 g CaCl_2_•2 H_2_O; filled to 150 mL ddH_2_O; pH adjusted to 7.5], 100 mg Collagenase II, 50 mL Stop Solution (400 mL PBS plus 100 mL FCS)) for 20 min. After resuspension of the digested liver in PBS/FCS the Hepatocytes (HC) were precipitated by centrifugation at 100 g (4 °C) for 10 min. The supernatant was centrifuged further at 500× *g* for 10 min and the pellet was transferred to an Optiprep gradient. The cells contained in the resulting upper phase were directly plated and after several passages stable fibroblast lines were established (Appendix A). MF were plated and cultured in DMEM containing 10% FCS for up to 6 passages. The interphase contained several other non-parenchymal cell types and was carefully collected. KC were separated using CD11b micro beads and for LSEC CD146 micro beads were used in magnetic separation procedures (all beads and columns were from Miltenyi Biotec). KC, HC and LSEC were directly lysed for PCR analyses. HSC were isolated as described previously [15] and were initially plated and cultured for a few days. HSC were then lysed for PCR without prior passaging.

### 2.1. Purification and Culture of Monocytes from Human Blood

Buffy coats were purchased from Heidelberg’s blood transfusion service (“Deutsches Rotes Kreuz”). A gradient was created by combining 15 mL Ficoll (density 1.077 ± 0.001 g/mL) and blood. It was centrifuged for 30 min (400× *g*), the interphase was collected and cells were washed three times with PBS. In the case of remaining red blood cells, an erythrolysis-buffer containing 1.5 M NH4Cl, 0.1 M NaHCO3, 0.01 M EDTA disodium salt was added and incubated for 5 min. Cells were resuspended in Roswell Park Memorial Institut (RPMI) medium containing 4.5 g/L glucose, 2.383 g/L HEPES buffer, 1.5 g/L sodium bicarbonate and 0.1% PLS but without FCS and plated (12-well-plate: 2 × 10^6^ cells/well). One hour later, non-adherent cells were washed away and medium was replaced by RPMI (as above) supplemented with 10% FCS. Cells were then cultured for 5–7 days to differentiate them into macrophages. For the experiments MP were serum-starved with medium containing only 2% FCS for two hours followed by stimulation with recombinant human BMP-9 (5 ng/mL; purchased from PeproTech Hamburg). After 24 h, LPS was added to some cultures and the cells were harvested 1 h and 24 h thereafter.

### 2.2. Cell Culture (LSEC, MF and J774A.1 Cells)

Upcyte^®^ LSEC are non-malignant cells derived from primary human LSEC. The cells were purcased from upcyte^®^ technologies, Hamburg and were cultured in Human Endothelial SFM (serum-free medium) purchased from Life Technologies.

MF were cultured in Dulbecco’s Modified Eagle Medium (DMEM) (Gibco, Thermofisher Scientific, Waltham, MA, USA) with 10% foetal calf serum (FCS) (Gibco, Thermofisher Scientific, Waltham, MA, USA), 0.1% L-glutamine (L-Gln) (Invitrogen, Germany) and 0.1% penicillin/streptomycin (PLS) (Biochrom, Germany). Cells were cultured at 37 °C in a 5% CO_2_ atmosphere. The murine cell line J774A.1 was cultured in RPMI supplemented with 10% FCS and medium was changed every 2nd day. Before stimulation the cells were serum-starved (0.5% FCS) for 2 h.

### 2.3. Affymetrix Arrays

Gene expression profiling was performed using arrays of Clariom^TM^ S Human Arrays (Thermo Fisher Scientific, Waltham, MA, USA). Biotinylated antisense cDNA was then prepared according to the standard labelling protocol with the GeneChip^®^ WT Plus Reagent Kit and the GeneChip^®^ Hybridization, Wash and Stain Kit (both from Thermo Fisher Scientific, Waltham, MA, USA). Afterwards, the hybridization on the chip was performed on a GeneChip Hybridization oven 640, then dyed in the GeneChip Fluidics Station 450, and thereafter scanned with a GeneChip Scanner 3000. All of the equipment used was from the Affymetrix-Company (Affymetrix, High Wycombe, UK). After normalizing the data by Robust Multichip Average algorithm (RMA) [16], multiple probes representing the same symbol were averaged for each array separately. Differential expressions were calculated as described by the limma package [17].

### 2.4. RT-qPCR

Total RNA was extracted from cells and tissues using the peqGOLD Total RNA purification kit (Peqlab) according to the manufacturer’s instructions. RNA was reverse transcribed into cDNA using the SensiFAST^TM^ cDNA Synthesis Kit (Bioline, London, UK). Real time quantitative PCR (RT-qPCR) was performed according to Analytik Jena innuMIX qPCR Master mix SyGreen protocol (Jena, Germany). Primers used for RT-qPCR are shown in Table 1. mRNA expression was normalized using suitable housekeeping genes as indicated in Table 1 and in the figure legends.

### 2.5. Primary Antibodies

For Western blots, a monoclonal rabbit anti-Smad3 (phospho S423/S425) antibody (EP823Y) (ab52903) was used that detects phosphorylated Smad1 and 3 (Abcam). To analyze the activation of the NfκB pathway, phosphorylation of p65 was detected using a monoclonal rabbit antibody to pp65 (D14E12) from Cell Signaling. For loading control, a monoclonal mouse anti-β-actin was used (Santa Cruz Biotechnology).

### 2.6. Preparation of Protein Lysates and Western Blot Immunoblotting

Total proteins from cell lysates were extracted on ice using RIPA lysis buffer (1x Tris-buffer saline, 1% Nonidet P40, 0.5% sodium deoxycholate, and 0.1% sodium dodecylsulfate) in the presence of protease and phosphatase inhibitors (Roche, Mannheim, Germany). Protein concentration was determined using a BioRad protein quantification assay kit (BioRad, Munich, Germany). Thirty micrograms of the protein extract were separated by 4%–12% SDS-PAGE (4–12% Bis-Tris Gel, NuPAGE, Invitrogen, Carlsbad, CA, USA) and transferred to nitrocellulose membranes (Pierce, Rockford, IL, USA). Nonspecific antibody binding was blocked with 5% non-fat milk in Tris-buffered saline containing 0.05% Tween 20 for 1 h. The membrane was incubated with the primary antibodies at 4 °C overnight. Horseradish peroxidase-linked goat anti-mouse and anti-rabbit secondary antibodies were used (1:5000; Santa Cruz Biotechnology, Heidelberg, Germany). Antibody detection was performed using a the Supersignal West Dura extended duration substrate (Thermofisher Scientific, Waltham, MA, USA) and chemiluminescence was detected with a Fusion solo chemiluminescence detection system (Vilber lourmat). Densitometric analyses were performed using ImageJ-Win64 software.

### 2.7. Mouse Experiments

All experimental animal protocols were performed in accordance with European Community policies, and approved by local committees (AZ: 35-9185.81/G-86/16). To analyze BMP-9/LPS effects in vivo C57/Bl6 mice were used (n = 5 per group). Mice received one intraperitoneal injection of either PBS (control), lipopolysaccharide (LPS, from Escherichia coli 0111:B4; Sigma Aldrich) (25 µg/mouse), BMP-9 (5566-BP-10/CF; R&D Systems) (100 ng/mouse) or a mixture of both. Mice were sacrificed at 2 h and 12 h after injections and livers were processed for RT-qPCR.

### 2.8. Immunohistochemistry

Immunhistochemistry was performed using the Leica BondMax System. Dilution of F4/80 antibody (clone: CI:A3-1; Biorad, Hercules, CA, USA) was 1:50. Pretreatment with Citrate buffer was performed for 20 min at room temperature. For quantification of the staining intensities 3 randomly chosen areas per specimen were photographed with 200-fold magnification. Red colour was quantified using ImageJ-Win64 software and results are presented as average values per treatment group +/−SD.

### 2.9. ELISA Measurements

10 mg frozen liver tissue (mouse) was homogenized in 500 µL RIPA buffer containing a protease inhibitor cocktail from abcam^®^ using an ULTRA TURAX (IKA 10T basic). Lysates were centrifuged (5′ at 10,000× *g*, at 4 °C) and the protein concentration of the supernatants was measured using the DC Protein Assay Reagents from BIO-RAD. Samples were further processed according to the manufacturer’s instructions. All ELISA Kits were purchased from abcam^®^ (IL-6: ab100713; IL-1 beta: ab100705; IL-10: ab100697).

### 2.10. Statistical Analysis

To analyze statistical differences (in GraphPad Prism 7.04) between groups of n = 3 the non-parametric ANOVA (Kruskal-Wallis) test was used and for n > 3 the non-parametric t-test (Mann-Whitney). Errors are the standard deviation (SD). *p* < 0.05 was considered statistically significant (*). ** indicates *p* < 0.01.

## 3. Results

### 3.1. LSEC and Kupffer Cells (KC) Express Alk1 and TLR4

In order to better understand how the presence of BMP-9 might affect the hepatic responses to LPS, we first analysed which liver cell types are expressing the LPS-receptor TLR4 and the high-affinity BMP-9 receptor Alk1. We showed previously that while BMP-9 itself seems selectively expressed in HSC, at least in mice, Alk1 was mainly expressed in LSEC and Kupffer cells (KC) [5]. Alk2, an alternative BMP-9 receptor with lower affinity, was basically expressed in all four cell types in mouse livers [5].

We used the same set of samples derived from primary isolated mouse liver cells and found that similar to Alk1, TLR-4 is not only expressed in KC but also in LSEC. In HSC, expression was lower, and in HC, it was almost not detectable (Figure 1A). In freshly isolated human liver cells; blood-derived macrophages (MP) and primary cultured HSC we could confirm that also in human liver, BMP-9 itself is mainly expressed in HSC (Figure 1B). TLR-4 expression in human tissue was also similar to mouse (Figure 1C) and Alk1 expression was principally expressed in all non-parenchymal liver cell types as well as in MP (Figure 1D). Alk2, another BMP-9 receptor with lower affinity than Alk1, seems to be more or less ubiquitously expressed in human liver (Appendix A). To confirm high purity of the individual cell types we additionally analyzed the expression levels of CD68 (a macrophage marker), Albumin (an HC marker) and CD34 (an endothelial cell marker) and αSMA (a myofibroblast marker) (Appendix A). We conclude that LSEC as well as KC/MP should be principally responsive to both BMP-9 as well as LPS.

### 3.2. Factors Secreted from LSEC, Including IL-6, Induce BMP-9 Expression in MF

Since BMP-9 itself is produced mainly in hepatic stellate cells (HSC) [5] (Figure 1B) and cross-talk between LSEC and HSC has been described [18], we next addressed the question of whether responses of LSEC to LPS might directly affect the neighbouring HSC or other fibroblastic liver cells. For this purpose, we used human upcyte^®^ LSEC and collected the conditioned medium from control- versus LPS-treated cells. We exposed primary human liver myofibroblasts (MF) that had been initially isolated from liver resection samples of patients to this medium and analyzed RNA expression changes. While classical fibrogenic markers like alpha-smooth-muscle actin (αSMA) or collagen type I (Col1a1) were not induced (Col1a1 was even significantly reduced), expression of BMP-9 itself was upregulated by the medium of LPS-treated LSEC (Figure 2A). These results imply that LPS initiates the secretion of factor(s) from LSEC that may not directly enhance fibrogenic activation of the cells but does cause increased expression of BMP-9. Since LPS led to a strong induction of IL-6 in LSEC (Figure 3A), we tested if IL-6 induces BMP-9 expression in MF. As shown in Figure 2B, this was indeed the case.

### 3.3. BMP-9 Antagonizes LPS Signaling in LSEC

In order to investigate if high levels of BMP-9 would in turn affect LSEC responses to LPS, we pre-treated LSEC with BMP-9 and subsequently stimulated them with LPS. On the RNA level we found that induction of certain pro-inflammatory genes like IL-6, CD54 (=ICAM1) and CXCR7 was significantly reduced by pre-treatment with BMP-9 (Figure 3A). By Western blot analyses, we further revealed that LPS-mediated activation of the NfκB pathway was dampened by pre-treatment of the cells with BMP-9 (Figure 3B). Otherwise, BMP-9 mediated activation of the Smad1-pathway or induction of the classical BMP-target gene, Id1, were not affected by additional LPS stimulation (Appendix A).

To get a more comprehensive picture of the gene expression patterns regulated by BMP-9 and LPS in LSEC, we performed Affymetrix microarray analyses (Figure 4). The results confirm that LPS-mediated upregulation of some inflammatory genes is reduced in the presence of BMP-9. Interestingly, Vcam1, an endothelial adhesion molecule/immunoglobulin superfamily member that was reported to induce lymphocyte adhesion to liver endothelium [19], was equally induced by BMP-9 as well as LPS.

Furthermore, BMP-9 seems to induce a state of capillarization by upregulating basal membrane components like fibronectin and collagen IV and reducing markers of differentiation like lyve1, stabilin1 and mrc1. BMP-9 also regulated the expression of several genes involved in angiogenesis and induction of fibrogenic TGF-β signalling (Figure 4). Most of these latter expressions were rather not changed by additional stimulation of the cells with LPS.

However, the regulation of target genes towards capillarization was often not significant (Figure 4, down-regulation of LSEC markers and upregulation of basal membrane proteins). We therefore performed additional expression analyses for lyve1 and col4a1 by regular PCR. The results confirm the overall impression that the pure BMP-9 effects are rather mild but are synergistically enhanced by co-stimulation with LPS (Appendix A).

### 3.4. BMP-9 Modulates Macrophages Responses to LPS

In the liver LSEC are not only in direct contact to HSC/fibroblasts on the one side but also to resident hepatic macrophages (Kupffer cells, KC) on the other side. As mentioned above, KC also express both, the BMP-9 receptor Alk1 as well as the LPS receptor TLR4. Therefore, we next investigated if the increased secretion of BMP-9 might as well act back on the LPS responses of macrophages. Because there are not many well characterized human macrophage cell lines available and the isolation of sufficient numbers of primary KC from human liver tissue was not realizable, we performed some experiments with the established murine cell line J774A.1, which had been described to be very responsive to LPS [20], and then additionally used primary human monocytes isolated from the blood of healthy donors. The latter were differentiated into macrophages (MP) by 5-day culture in serum-containing medium. We found that LPS reduced basal and BMP-9-mediated induction of Id1 or hepcidin (hamp) expression to different degrees depending on the macrophage cell type and incubation condition and it led to reduced presence of phosphorylated Smad-1 (Figure 5A,B). Unexpectedly, although LPS reduced Id1 expression already after 1 h, the reduced phosphorylation of Smad1 was only detectable at 24 h of stimulation (Figure 5B). One explanation of this observation could be that BMP-9 induces Id1 not only via the Smad-pathway, but additionally via other pathways that are also (and perhaps more rapidly) inhibited by LPS.

Using the human macrophages we further detected a synergistic upregulation of IL-6 by co-treatment (Figure 5C). When looking at other inflammatory mediators (IL-10, TNFα, IL1β) the results again pointed to an enhanced pro-inflammatory expression pattern in those cells that had been treated with both, BMP-9 and LPS (Figure 5C).

### 3.5. Effects of BMP-9 Plus LPS In Vivo in Mice

Having found that BMP-9 seems to directly interfere with hepatic responses to LPS in human cells in vitro we finally investigated such responses in vivo in mice. For this purpose, we intraperitoneally injected BMP-9, LPS or both, sacrificed the animals after 2 h (early responses) and 12 h (late responses) and isolated RNA from liver tissue samples. LPS-mediated induction of expression of the anti-inflammatory IL-10 was significantly enhanced by co-treatment at the early time-point but, similar to our results obtained with the human macrophages, after 12 h it was rather reduced. Induction of TNFα was significantly enhanced by co-treatment at the 2 h time-point and IL1β showed a similar tendency (Figure 6A). As expected, the expression of IL-6 was very strongly induced in these livers by LPS already at the early time-point, however, we could not observe any further upregulation by LPS plus BMP-9 (Figure 6A). On the protein level we could confirm most of these tendencies by performing ELISA measurements; however, here we did not reach statistical significance (Figure 6B). To address the question whether the numbers of macrophages in these livers were changed we performed immunohistochemical stainings against the macrophage marker F4/80. The results presented in Figure 6C demonstrate that while LPS, as expected, led to increased presence of F4/80-positive cells, such increase was already observed by single BMP-9 treatment alone. Co-treatment did not point to any further increase.

## 4. Discussion

We have previously shown that high levels of BMP-9, produced by hepatic stellate cells (HSC), promote liver fibrosis and counter-act hepatic regenerative processes in C57BL/6 mice [5]. Since we did not observe any direct HSC-activating properties of BMP-9 on isolated HSC in vitro, the underlying cellular mechanism of this pro-fibrogenic action of BMP-9 remained largely unknown. Recently published data even point to the conclusion that depending on the genetic background, BMP-9 may even act anti-fibrogenic at least in murine liver [21,22]. The possible functions of BMP-9 in human hepatic regeneration/wound-healing processes remain to be investigated. In the present work, we aimed at testing the hypothesis that possible adverse (e.g., pro-fibrogenic) actions of BMP-9 in the liver might be the result of cellular cross-talk between HSC (or other liver fibroblasts), LSEC, and macrophages. LPS was applied because it represents a very well-studied initial mediator of liver damage and for translational purposes, we used mainly human cells to investigate how BMP-9 expression and action are changed in response to LPS stimulation in these cell types.

The BMP-9 type I receptor with highest affinity is Alk1 [23] and LPS transmits its signals mainly via TLR4 [24]. By RT-qPCR using isolated liver cells we found that both, LSEC as well as macrophages (MP) express comparably high amounts of both receptors (Figure 1). We confirmed our earlier results with mouse cells and showed that indeed also in human cells HSC are the cell type with highest basal BMP-9 expression (Figure 1B) and these cells also expressed both receptors. Human monocyte-derived MP isolated from the blood of healthy donors expressed levels that were comparable to those detected in KC. Human HC expressed again very low levels of both, at least when directly compared to LSEC and KC/MP; however, more or less all cell types analyzed expressed basal levels of Alk2, an alternative BMP-9 receptor with lower affinity than Alk1. Expression of TLR4 in KC and their responsiveness to LPS has already been reported for primary rat and mouse KC [25,26], as well as murine LSEC [27]. Data obtained with primary human cells, however, are less frequently reported.

After intraperitoneally injecting LPS into mice, NfκB activation in HC as well as in desmin-negative non-parenchymal cells was described to depend upon the presence of KC, whereas that of HSC was not [11]. The authors concluded that NfκB activation in HC and desmin-negative non-parenchymal cells (like LSEC) occurred indirectly by LPS-triggered secretion of factors like TNFα from KC. HSC in contrast did not need the presence of KC for this response. These data obtained with murine models already imply that LPS responses are highly dependent on the individual cross-talk between the different liver cell types. How BMP-9 might affect such cross-talk in the context of LPS is not fully understood yet. However, it has been described that BMP-9 increases neutrophil recruitment to LPS-stimulated human aortic pulmonary and blood outgrowth endothelial cells (HPAEC and BOEC, respectively) in an Alk1-dependent manner [28]. In line with these data we now show that 12 h after injection of BMP-9 to mice, the amount of F4/80-positive cells in the liver significantly increased to a similar degree as was achieved by injection of LPS (Figure 6C). These data support the conclusion that BMP-9 indeed modulates invasion of inflammatory cells into the liver in vivo.

Our previous results showed that direct exposition of HSC with LPS in vitro down-regulates BMP-9 expression [5]. However, if this also happens in vivo, and how BMP-9 expression is in general regulated in the liver, is not known. HSC are not the only fibroblastic cell type of the liver and other populations like portal fibroblasts or mesenchymal stem cells (MSC) exist [29,30]. These other fibroblastic cell types might as well become pro-fibrogenic myofibroblasts [31] and could well be sources of BMP-9. In vivo LPS from the blood stream should first act on LSEC and KC/MP and rather secondarily on the cells in the Space of Disse. Therefore, we aimed at investigating, if there exist secondary mechanisms, how LPS might control BMP-9 production. For this purpose, we first stimulated LSEC with LPS, collected the conditioned medium from these cultures and stimulated liver myofibroblasts (MF; see Appendix A for characterization of these cells) with it. Obviously the secretom of LPS-stimulated LSEC contained BMP-9-inducing factors (Figure 2A). Because it has been documented that LPS induces the expression of IL-6 in HUVEC [32] as well as in HPAEC [33] and murine LSEC [27], we tested if direct stimulation of MF with IL-6 would also induce BMP-9 expression and this was indeed the case (Figure 2B), demonstrating that IL-6 might be one important regulator of BMP-9 expression levels in the liver.

Direct exposition of HSC to LPS reduces BMP-9 expression but LPS-mediated secretion of IL-6 from adjacent cells induces it at least in MF. We think that this seeming discrepancy of LPS effects can be explained by the different locations and functions of LSEC and HSC/MF in the injured liver. Acute damage, e.g., a single injection of CCl4, most likely leads to an initial destruction of parts of the endothelial cell linings of the sinusoids. This allows LPS to directly hit the HSC but in this situation BMP-9 would interfere with regeneration (e.g., by blocking proliferative responses of the parenchyme) therefore in HSC direct exposition to LPS induces down-regulation of BMP-9 expression. Later, when LSEC have started to recover and the gaps in the sinusoidal walls are closing, HSC are not directly exposed to LPS any more. In parallel, in order to complete the regenerative response and aid in maturation of the endothelium, BMP-9 is needed again and now LPS acting on LSEC re-stimulates BMP-9 production via secretion of IL-6. This concept fits with our previous finding that a single injection of LPS to mice initially, at a time-point where LPS concentration is so high that it can also reach the HSC, leads to a strong decrease of hepatic BMP-9 expression which is followed by a longer phase of enhanced expression [5]. It should be noted, however, that in contrast to HSC, the MF used here did not express any basal levels of BMP-9 (Appendix A) and should therefore not be involved in physiological/homeostatic BMP-9 effects.

Interestingly we found that BMP-9 in turn directly antagonizes LPS-mediated expression of IL-6 in LSEC (Figure 3A) thereby directly regulating its own expression in terms of a negative feed-back loop. Such antagonistic effects of BMP-9 were also found regarding several members of the Cxcl/Ccl familiy of cytokines (Figure 4) and might be the result of a direct inhibition of the NFκB pathway by BMP-9 (Figure 3B). However, induction of other members of these families like Cxcl3, 5, and 8 for example, was not antagonized and Vcam1, a mediator of lymphocyte adhesion to the endothelium [19], was even synergistically upregulated by BMP-9 and LPS. The latter synergistic effect of BMP-9 and LPS on Vcam1 expression had previously been described in HPAEC and BOEC [28] and fits well with our assumption that BMP-9 enhances the invasion of inflammatory cells into the liver.

One of the first events during liver fibrogenesis is that LSEC lose their specific phenotype. They close the inter-cellular gaps, lose their fenestrae and start to produce basement membrane proteins. This process is termed capillarization. If LSEC become capillarized they lose their protective properties like phagocytosis and keeping HSC unactivated [18,34]. In cirrhosis, LSEC frequently transform into a vascular type with a basement membrane, which interferes with the bidirectional exchange of molecules. Several growth factors, including vascular endothelial growth factor (VEGF), regulate the state of transformation of LSEC into the vascular phenotype. Our results point to the conclusion that at least in human LSEC also BMP-9 seems to induce some features of capillarization like down-regulation of lyve-1 and upregulation of basement membrane proteins like fibronectin and collagen type IV (Figure 4 and Appendix A). In addition BMP-9 induced expression of TGF-β2 as well as several activators of latent TGF-β, like thrombospondin (thbs1), ecm1 and integrin αv. Further, the expression of the pro-proliferative cytokine PDGF-B was enhanced, all together pointing at pro-fibrogenic effects of BMP-9 in these human cells. It must be pointed out however that the effects of BMP-9 alone towards capillarization of LSEC were rather mild and only became prominent in the context of co-stimulation with LPS.

Regarding the effects of BMP-9 on macrophages (MP), we first observed that different from the results obtained with LSEC, in MP induction of classical BMP-9 target genes like Id1 and hamp (hepcidine) was reduced by additional stimulation with LPS (Figure 5A,B). Accordingly, at least after 24 h of LPS stimulation, activation of the BMP-Smad pathway was also reduced, implying that there exists some kind of cell-type specific interference between the BMP-9 and LPS pathways in MP. Whether this is due to direct interference with Smad-phosphorylation or is rather the result of a negative feed-back on Smad1 expression itself remains to be investigated. Also regarding regulation of expression of IL-6 and diverse inflammatory mediators, the effects of co-treatment with BMP-9 and LPS were different than in LSEC, pointing to a pro-inflammatory synergistic action (Figure 5C). At least for some of these mediators a similar trend was observed using liver samples from BMP-9/LPS treated mice (C57/Bl6; Figure 6). It remains to be investigated if such effects indeed occur in humans as well and if human KC react similar to BMP-9 as monocyte-derived MP.

In conclusion, it seems that the level of BMP-9 expression in fibroblastic liver cell populations needs to be tightly controlled and that depending on the given situation and cellular context, too much as well as too little BMP-9 may both result in disturbed regeneration or maturation, respectively. Thereby, a disturbed fine-tuning of the cellular cross-talk that regulates BMP-9 expression might result in enhanced or inhibited inflammation and fibrosis in the liver.

## Figures and Tables

**Figure 1 cells-09-00617-f001:**
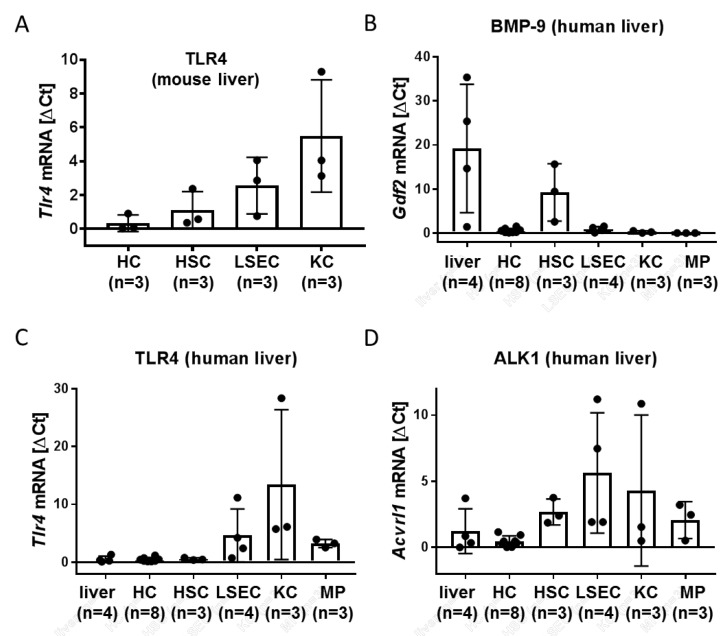
Expression of BMP-9 and LPS receptors in different liver cell types. (**A**) Four different liver cells types (hepatocytes (HCs), HSCs, liver sinusoidal endothelial cells (LSECs) and Kupffer cells (KCs)) were simultaneously isolated from healthy mouse livers, directly lysed and total RNA was purified. Tlr4 expression levels were determined by RT-qPCR. Data were normalized to the average Ct value of each individual experiment (isolation): Ct-target gene minus Ct-average. Results are presented ± SD from n = 3 isolations. (**B**–**D**) HSC, KC, HC or LSEC were isolated from human resected (non-malignant) liver tissue from the surrounding of colorectal cancer metastases. KC, HC and LSEC were directly lysed for RNA purification, HSC were cultured for a few days. Macrophages (MP) were generated from monocytes isolated from the blood of healthy donors by 1-week culture in serum-containing medium. For comparison RNA samples of whole liver tissue were additionally analyzed (“liver”). Expression levels of Bmp-9 (**B**) Tlr4 (**C**) and Alk1 (**D**) were determined by RT-qPCR and normalized as in (**A**). Data are expressed as average values ± SD (ΔCt of target gene versus the average Ct of all samples). Each dot represents an individual cell isolation.

**Figure 2 cells-09-00617-f002:**
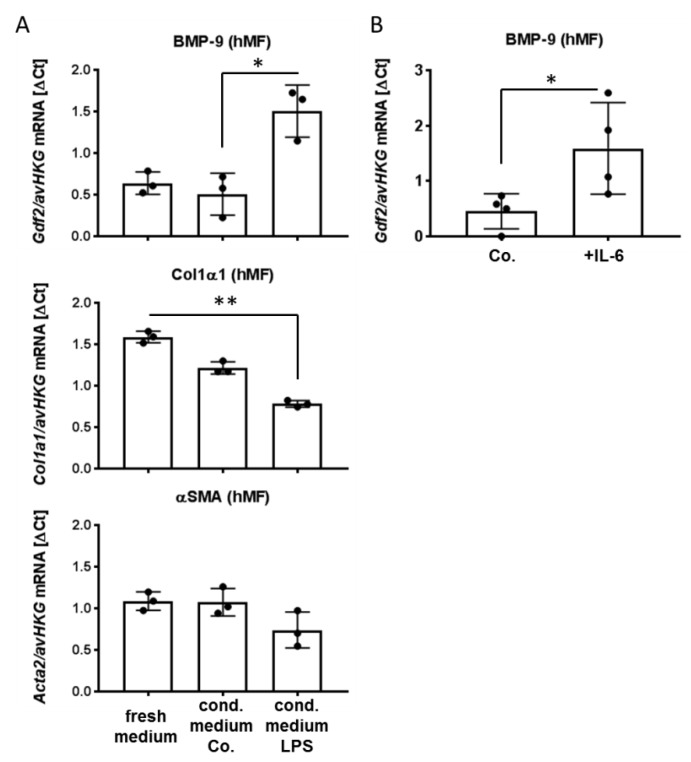
BMP-9 expression in human liver myofibroblasts (MF) is enhanced by IL-6. (**A**) LSEC were cultured for 24 h in the presence or absence of LPS (100 ng/mL) and the conditioned medium was collected. Human liver MF were cultured either in a 1:1 mixture of fresh LSEC-medium and DMEM respectively (“fresh medium”), or in a 1:1 mixture of DMEM and conditioned medium from unstimulated (“cond. medium Co.”) or LPS-stimulated LSEC (“cond. medium LPS”). BMP-9, Col1a1 and αSMA expression levels were determined by RT-qPCR. Data are expressed as average values ± SD (ΔCt of target gene versus average of two house-keeping genes: b2M and Rpl19; “avHKG”). (**B**) Human liver MF were stimulated with IL-6 (1 ng/mL) for 1 h and BMP-9 expression was determined by RT-qPCR as in (**A**). *p* < 0.05 was considered statistically significant (*). ** indicates *p* < 0.01. Each dot represents an individual sample’s value.

**Figure 3 cells-09-00617-f003:**
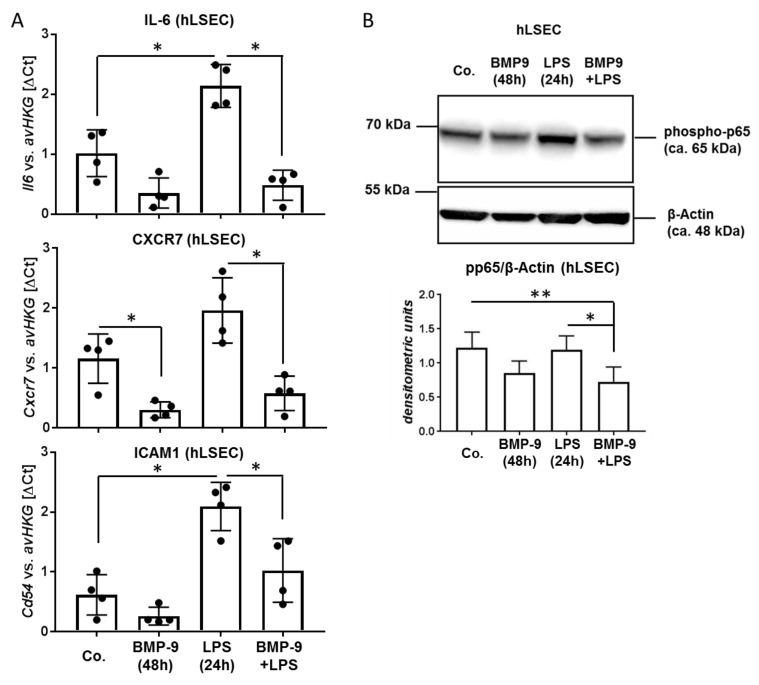
In human LSEC pre-treatment with BMP-9 antagonizes LPS-mediated activation of the NfκB pathway. (**A**) Human LSEC were pre-cultured with or without BMP-9 (5 ng/mL) for 24 h followed by addition of LPS (100 ng/mL) for another 24 h as indicated. Expression levels of IL-6, CXCR7 and CD54 were determined by RT-qPCR. Data are expressed as average values±SD (ΔCt of target gene versus house-keeping gene). (**B**) Using the same experimental setup as in (**A**) protein lysates of LSEC were subjected to Western blot analysis. Activation of the NfκB pathway was determined by detection of phosphorylated p65 and β-actin was used as loading control. One representative blot plus densitometric analyses results of n = 3 is shown. *p* < 0.05 was considered statistically significant (*). Each dot represents an individual sample’s value.

**Figure 4 cells-09-00617-f004:**
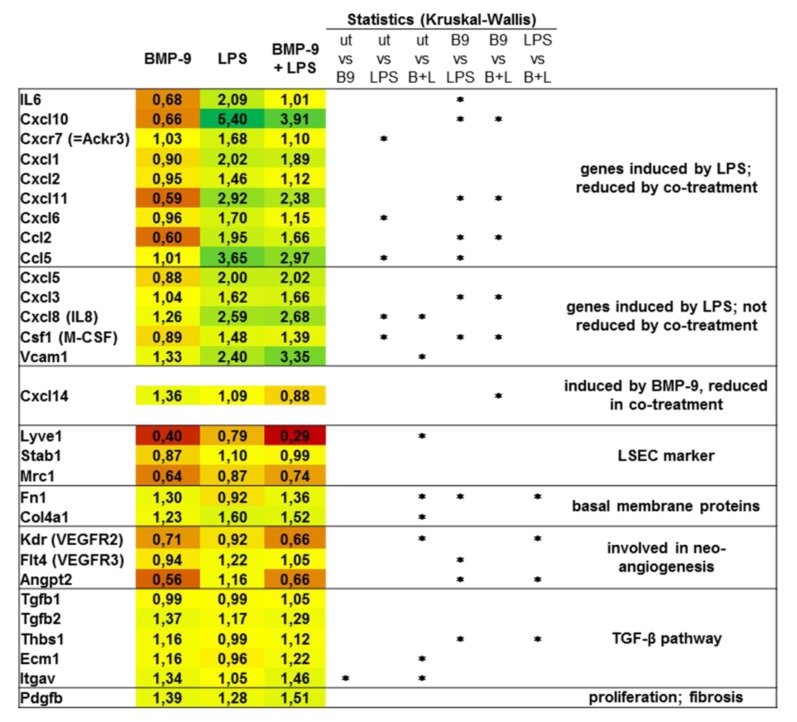
Affymetrix array analysis of BMP-9/LPS responses in human LSEC. Human LSEC were stimulated with BMP-9 and LPS as in Figure 3 and RNA samples from 3 independent experiments were subjected to Affymetrix array analysis. Data for selected targets are presented as average expressions expressed as “fold of untreated”. Red colour indicates down-regulated expression and green upregulation. Bright yellow was set as 1 (= unchanged expression compared to untreated). *p* < 0.05 was considered statistically significant (*).

**Figure 5 cells-09-00617-f005:**
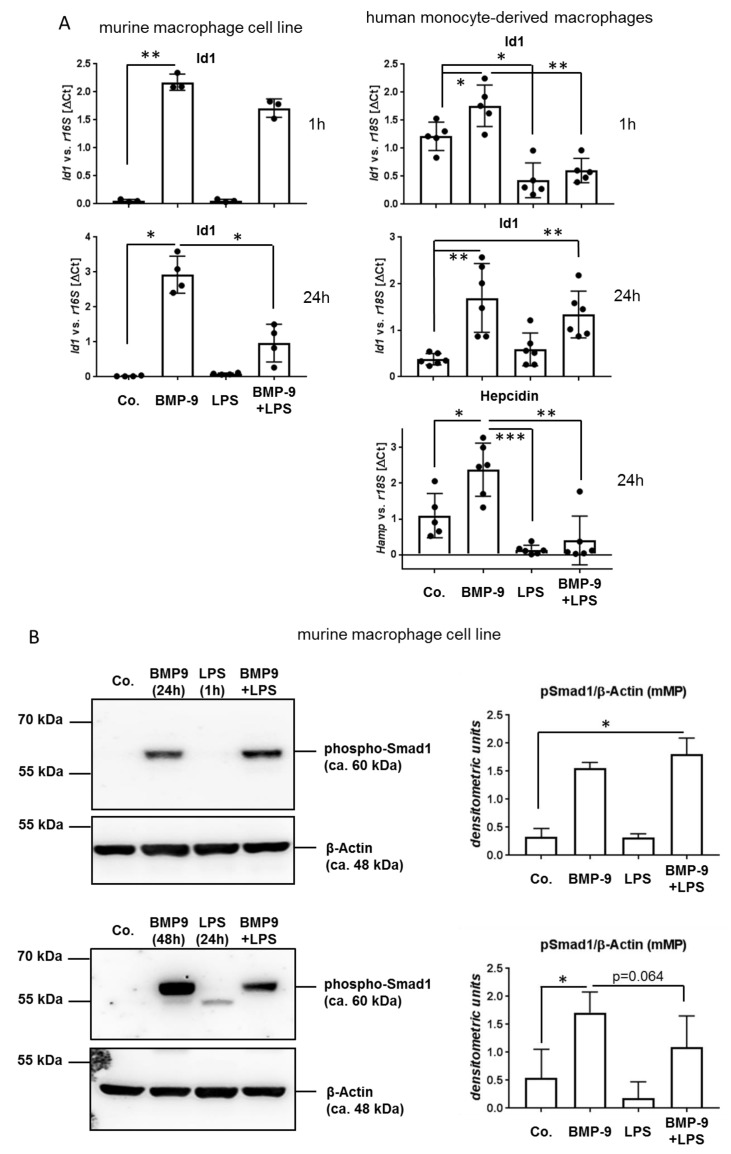
In macrophages stimulation with LPS plus BMP-9 reduces activation of the classical BMP/Smad-pathway but enhances inflammatory targets. (**A**) Cultured macrophages were stimulated with BMP-9 and LPS using the same experimental setup as for LSEC (Figure 3) (“24 h”) plus an additional setup with only 1 h of LPS stimulation (“1 h”). We used either a murine macrophage cell line (J774A.1; left hand graphs) or primary human monocyte-derived macrophages obtained from the blood of healthy donors (right hand graphs). Expression levels of the BMP-9 target genes Id1 and hamp (= hepcidin) were determined by RT-qPCR. Data are expressed as average values ± SD (ΔCt of target gene versus house-keeping gene). (**B**) Using the same experimental setup as in (**A**) protein lysates of J774A.1 cells were subjected to Western blot analysis. Activation of the Smad1/5/8 pathway was determined by detection of phosphorylated Smad1 and β-actin was used as loading control. One representative blot plus densitometric analyses results of n = 3–4 is shown. (**C**) In the same human macrophage samples as used in (**A**) we further analyzed the expression levels of the inflammatory mediators IL-6, IL1β, TNFα and IL10 by RT-qPCR. *p* < 0.05 was considered statistically significant (*). ** indicates *p* < 0.01. Each dot represents an individual sample’s value.

**Figure 6 cells-09-00617-f006:**
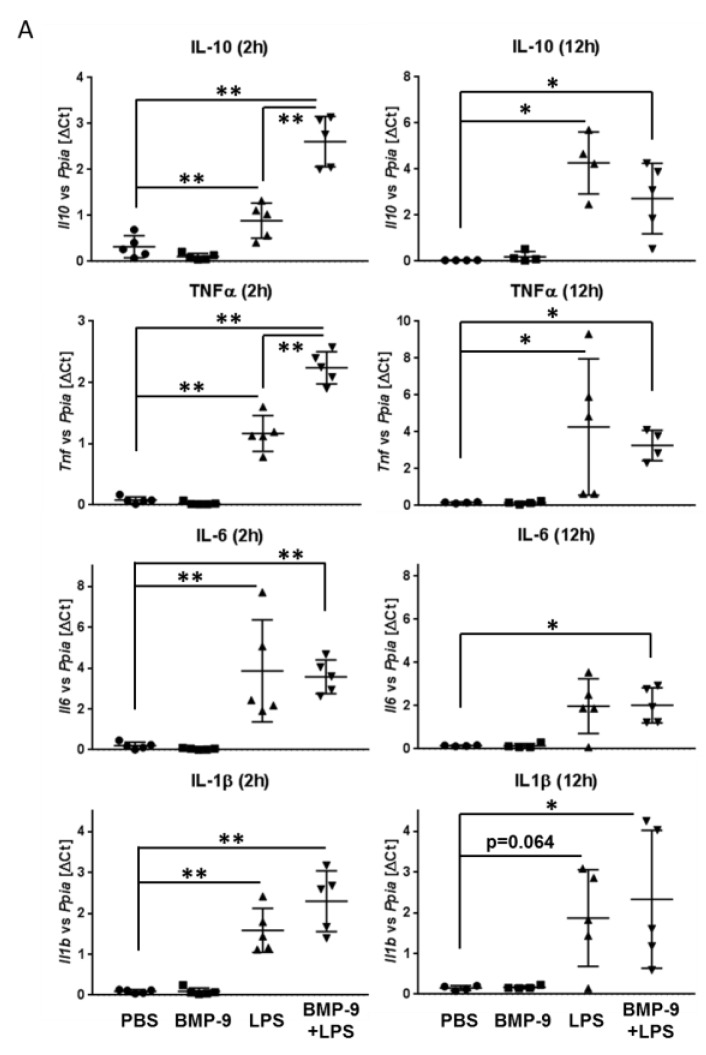
Injection of BMP-9/LPS into mice results in enhanced LPS-mediated induction of pro-inflammatory targets. (**A**) RNA samples were prepared from the livers of mice (C57BL/6; n = 5 per group) 2 h and 12 h after injections of PBS (control), BMP-9 (100 ng/mouse), LPS (25 µg/mouse) or a mixture of both and expression levels of IL-10, IL-6, IL-1β and TNFα were determined by RT-qPCR. Data are expressed as average values ± SD (ΔCt of target gene versus house-keeping gene (Ppia)); (**B**) Liver tissue was processed for ELISA measurements of protein levels of IL-6, IL-1β and IL-10; (**C**) Immunohistochemistry was performed to analyse the amount of F4/80-positive cells (only 12 h time-point). Representative staining examples are shown in the upper part (200-fold magnification) and quantification of the staining intensities is shown below. *p* < 0.05 was considered statistically significant (*). ** indicates *p* < 0.01. Each dot in (**A**) and (**B**) represents the average value of samples from the livers of an individual mouse. Dots in (**C**) represent densitometric values of individual image sections.

**Table 1 cells-09-00617-t001:** Sequences of the primers used for RT-qPCR (5′ → 3′ orientation).

Acta2 (αSMA)	human	CCCTGAAGTACCCGATAGAAC	GGCAACACGAAGCTCATT
Acvr1 (Alk2)	human	TCTCTGTAGTGTTCGCAGTATGT	CGTTCTTGGTTGCGCCTTTT
Acvrl1 (Alk1)	human	CATCGCCTCAGACATGACCTC	GTTTGCCCTGTGTACCGAAGA
Alb	human	CTTGAATGTGCTGATGACAGG	GCAAGTCAGCAGGCATCTCAT
Cd271 (NGFR)	human	AACCTCATCCCTGTCTATTG	GTTGGCTCCTTGCTTGTT
Cd34	human	CTACAACACCTAGTACCCTTGGA	GGTGAACACTGTGCTGATTACA
Cd44	human	GGAGCAGCACTTCAGGAGGTTAC	GGAATGTGTCTTGGTCTCTGGTAGC
Cd54 (ICAM1)	human	ATGCCCAGACATCTGTGTCC	GGGGTCTCTATGCCCAACAA
Cd68	human	GCTACTGGCAGCCCCAGGG	GCTCTTGGTAGTCCTGTGG
Cd90	human	ATGAACCTGGCCATCAGCA	GTGTGCTCAGGCACCCC
Col1a1	human	GTGCGATGACGTGATCTGTGA	CGGTGGTTTCTTGGTCGGT
Col4a1	human	TGGTGACAAAGGACAAGCAG	TAAGCCGTCAACACCTTTGG
Cxcr7	human	CTATGACACGCACTGCTACATC	CTGCACGAGACTGACCACC
Fn1	human	CGGTGGCTGTCAGTCAAAG	AAACCTCGGCTTCCTCCATAA
Gdf2 (BMP9)	human	pre-designed sequence: QT00210462; Qiagen	
Grem1	human	TCATCAACCGCTTCTGTTACG	GGCTGTAGTTCAGGGCAGTT
Hamp	human	CTGACCAGTGGCTCTGTTTTC	GAAGTGGGTGTCTCGCCTC
Id1	mouse	CTTCAGGAGGCAAGAGGAAA	CAAACCCTCTACCCACTGGA
Id1	human	CTGCTCTACGACATGAACGG	GAAGGTCCCTGATGTAGTCGAT
Il10	human	TCAAGGCGCATGTGAACTCC	GATGTCAAACTCACTCATGGCT
Il10	mouse	CAGAGCCACATGCTCCTAGA	GTCCAGCTGGTCCTTTGTTT
Il1b	human	AGCTACGAATCTCCGACCAC	CGTTATCCCATGTGTCGAAGAA
Il1b	mouse	CTGCTGGTGTGTGACGTTCCCAT	GGTCCGACAGCACGAGGCTTT
Il6	human	ACTCACCTCTTCAGAACGAATTG	CCATCTTTGGAAGGTTCAGGTTG
Il6	mouse	TAGTCCTTCCTACCCCAATTTCC	TTGGTCCTTAGCCACTCCTTC
Lyve1	human	AGGCTCTTTGCGTGCAGAA	GGTTCGCCTTTTTGCTCACAA
Nnmt	human	GAGATCGTCGTCACTGACTACT	CACACACATAGGTCACCACTG
Tlr4	mouse	AAGAGCCGGAAGGTTATTGTG	CCCATTCCAGGTAGGTGTTTC
Tlr4	human	AGTTGATCTACCAAGCCTTGAGT	GCTGGTTGTCCCAAAATCACTTT
Tnf	human	CCTCTCTCTAATCAGCCCTCTG	GAGGACCTGGGAGTAGATGAG
Tnf	mouse	TCCCAGGTTCTCTTCAAGGGA	GGTGAGGAGCACGTAGTCGG

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
