# Peer review of "BMP-9 Modulates the Hepatic Responses to LPS"

_cells, 2020, doi:10.3390/cells9030617_

Round 1
Reviewer 1 Report
This article by H. Gaitantzi and collaborators studies the effect of BMP9 (bone morphogenetic protein-9) and lipopolysaccharide (LPS), a bacterial endototoxin, on liver sinusoidal endothelial cells (LSECs). This article follows a previous publication by the same team (Breitkopgf-Heinlein K. et al., Gut, 2017) that reported that overexpression of BMP9 in the liver is pro-fibrogenic. The present article reports several original findings and nicely complements the characterization of BMP9 biological action in the liver.
Using primary cultures of human LSECs and human HSCs (hepatic stellate cells), the authors show that LPS treatment of LSECs induces the secretion of IL6 (interleukin 6) and possibly other cytokines which in turn induce the synthesis and secretion of BMP9 by HSCs. This is an original observation since no regulator of BMP9 synthesis was described so far and since the results were obtained on human cells.
The authors then treated LSECs with BMP9, LPS or a combination of both. They subsequently analyzed the transcriptomic response of these endothelial cells and observed that several markers of differentiated LSECs (Lyve-1, Stabilin-1, Mrc-1) were down-regulated by BMP9, both in the absence and in the presence of LPS, suggesting a capillarization process. This is also an interesting observation although the capillarized phenotype was not complete (see comments thereafter).
The authors have also injected either BMP9, LPS or both to mice and analyzed the hepatic production of various cytokines. They report that IL-6, IL-10 and TNF-α are strongly induced by LPS as early as 2h-post treatment and that BMP9 only slightly modulates these responses.
Specific comments and questions :
Introduction :
page 2 line 5-6 : the authors should indicate that the affinity of BMP9 for ALK1 is much higher than that for ALK2. The two receptors cannot be considered equal.Figure 1 :
The authors should analyze the expression of ALK2 in the same cell types as they did for ALK1. 1A : How are the RT-qPCR values normalized for RNA amounts variations ? The representation of results as % of total is unappropriate because it does not make sense to add the mRNA expression levels from each cell type. It would be preferable to express the data as ratios to one cell type (e.g. LSECs). 1B : The unit (ΔCt) is different from that used in Figure 1A whereas the legend indicates a similar calculation. Please homogenize the representations, taking into consideration the comments above.Figure 2 :
Why are the results of Figure 2A and 2B normalized to different houskeeping genes ?Figure 3B :
As the Western blot was performed in 3 independent experiments, please show the mean±SEM of the quantitations of the results.Figure 4 and Results section page 8, line 5-6 :
The authors should report the p values of the differences shown as a supplementary table. I am not convinced that the inductions of Fn1 (x1.3) and Col4a1 (x1.23) are significant. They are very small and the comment on line 5&6 should be restricted to the effects on Lyve1, Stab1, Mrc1, which are more convincing. Therefore, the authors should describe a partial capillarization process.Figure 5 :
Figure 5 should be packed so that it fits in a single page. The long-term effects of BMP9+LPS on phospho-Smad1 are interesting but the authors should show the levels of total Smad-1 in order to be certain that the effect is on Smad1 phosphorylation rather than on Smad1 expression.Figure 6 :
The in vivo effects of BMP9 are relatively weak. This could be due to the fact that i.p. injection of BMP9 did not significantly raise the plasma levels of BMP9. The authors should report the circulating levels of BMP9 under all 4 conditions tested. The results shown in Supplementary Figure 2 (in particular IL6 expresion levels) should be combined in a single figure with those shown in Figure 6.Discussion :
7th paragraph : As mentioned previously, the comments on BMP9-induced capillarization should be tempered.General comments :
Was gene expression mesaured by RT-PCR or RT-qPCR ? Please use the same abbreviation in the text and in the Figure legends. Please add the number of repeats (n) of each experiment to the Figure legends. Statistical analysis was performed using a Student’s t-test, which seems unappropriate, given the low number of repeats (usually <5). A non-parametric test (Kruskall-Wallis) would be more appropriate. Please check again the significance of the results using a non-parametric test.
Reviewer 2 Report
This is an interesting article where the authors are investigating the adverse effects of BMP-9 and the cross talk been HSC, LSEC and macrophages. Importantly the authors have used human cells comparing it with mouse liver cells and finally a mouse model experiment. But the enthusiasm for this interesting work is dampened by mostly mRNA data for studying the effect of BMP-9. Thus the manuscript requires a major revision including protein data for all the cytokines studied.
A human subjects paragraph has to be included in the materials and method Phenotypic characterization using markers and functional assay specific to each cell type from human liver have to be included and the purity of each of the cell population has to be included in manuscript Statistical analysis for figure 1 is needed. Figure 1B and C should include HSC for comparison Additional data is needed whether BMP-9 is made by human hepatocytes unlike mice where HSC selectively secretes BMP-9. Given the anatomical location of HSC, which are perisinusoidal cells located in space of Disse, it is highly possible that the cellular cross talk involved hepatocytes LSEC and HSC and not KC or macrophages. Additional justification and data is needed to show that monocyte derived macrophages is similar to Kupffer cells. In figure 5 the basal level of cytokine mRNA is high in untreated for TNF-α and IL-10. More explanation is needed why this is the case. Conventional nomenclature have to be used for IL-6, IL-1b, TNF-α etc in the figures Additional protein data using ELISA is needed for the cytokines in Figure 5. Since mRNA is not always translated to protein. Most importantly protein data is needed in place of Figure 6.Author Response
please see attachment

Round 2
Reviewer 2 Report
Dear Authors,
Its very important that protein analysis be done for Figure 6. If you need more time please let us know
Author Response
Dear Reviewers and Editors,
thank you for your patience. We made the following changes to our manuscript:
Fig. 1:
Already in the first round of review, Reviewer#1 criticized that in our human cell type collection we did not include primary HSC and could therefore not show that also in humans (as in mice) BMP-9 is mainly expressed in HSC and not in the other cell types investigated here. Within the first review time-frame we did not manage to solve this problem but because we principally fully agreed with this criticism, we searched for a cooperation partner who might be in the position to help us out with primary human HSC. Luckily Claus Hellerbrand from the University of Erlangen was now able to supply the RNA from primary HSC of 3 donors. Therefore new Figure 1B now shows the missing BMP-9 expression in human HSC. A small disadvantage is the fact that these cells were not directly lysed after isolation but had been in culture for a few days. Therefore these cells can be considered “activated”. Nevertheless also activated cells, at least in mouse, should express high amounts of BMP-9 (as shown in our previous paper, Breitkopf-Heinlein et al., Gut 2017).
You might notice that the “n-numbers” in the new Figure 1 are smaller than before. With the aim to use only cell preparations of high purity we have performed extra rounds of PCRs against many different cell type markers (some of these results are presented in Supplementary Figure 2) and we included only those with highest purity (at least n=3 per cell type).
Fig. 2 and Supplementary Fig. S1:
All labeling “HSC” in Fig. 2 was replaced by “MF” which stands for “myofibroblasts”. This is for the following reason: as stated above, also activated HSC should express BMP-9. But when we tested the fibroblastic cells that were used in these experiments they basely did not express any BMP-9 (see Suppl. Fig. 1B). They were also negative for an additional HSC-specific marker, CD271 (NGF-receptor). How could this be? The cells had been isolated from human liver by densitiy gradient centrifugation and were derived from the fraction that should contain the HSC. However, the number of cells that we obtained in this way was very low and could only be increased by prolonged culturing (in serum-containing medium). Because the cells rapidly developed the typical morphology of activated HSC (= myofibroblast morphology) and expressed typical markers like aSMA, Collagen I or Fibronectin (Suppl. Fig. 1B), we initially considered them to be indeed HSC. After reading the literature carefully we now think that we have generated fibroblastic lines which most likely represent sub-types of liver myofibroblasts like mesenchymal stem cells or portal fibroblasts. They indeed express markers of such other fibroblastic types to varying levels (Suppl. Fig. 1B). It is beyond the scope of this manuscript to terminally characterize these cells but the finding that they upregulate BMP-9 under the conditions described here is still important. It even shows that HSC are not the only fibroblastic cell type of the liver that produces BMP-9 in humans. We included these thoughts in the discussion and cited further literature relevant to this topic.
Figure 6:
Regarding the mouse experiment where we injected BMP-9/LPS we did the following additional analyses: Our previous data on the RNA-level showed some interesting tendencies but also quite high variations (Fig. 6 in previous version). Therefore we went back to the frozen tissue pieces of these livers and prepared more RNA samples per each animal (only 2h-time-point). With these extra samples we repeated PCRs against all targets and made a final analysis of all samples (old plus new). This did indeed reduce the error and led to the fact that now inductions of IL-10 and TNFa were significantly higher in the co-treatment group (see Fig. 6A). In an attempt to support these findings on the protein level, we made lysates from these tissues and applied ELISAs for IL-10, IL-6 and IL1beta. Unfortunately the ELISA for TNFa would have had a delivery time of 6-8 weeks and could therefore not be used (if you decide that we should still include these data, within 2 months we could deliver them). These ELISA results show mild tendencies supporting the RNA results (Fig. 6B), but they did not reach statistical significance. This could be due to the rather small number of samples (n=5) or maybe we did not include the perfect time-point for the protein level differences.
With the aim to find out if the number of inflammatory cells in the livers of these mice differed in the different treatment groups, we searched for the help of a pathologist, Carolin Mogler, and performed immunohistochemical stainings against F4/80 (a macrophage marker). The results, presented in Fig. 6C, do not show more positive cells in co-treatment versus LPS alone but it does show already enhanced numbers of such cells in the BMP-9-only group. This is an interesting new finding which matches nicely with published in vitro data pointing to a role of BMP-9 in enhancing the passage of inflammatory cells through endothelial barriers (corresponding literature is cited as #28). It also matches with our finding that BMP-9 induces Vcam in LSEC (Fig. 4) and supports the conclusion that BMP-9 indeed modulates inflammatory actions in the liver at least in part by promoting infiltration of macrophages into the liver. These results are now included and discussed in the revised version.
In summary we believe that we were able to substantially improve the quality and novelty of this study. Of course we could not answer all questions and we are aware that we even produced some new questions but isn’t that the nature of science? We think that we can deliver a nice set of new details about BMP-9 functions in the liver that will help to better understand the role of this interesting cytokine in health and disease.
Hoping that you can share this assumption we remain awaiting your final decision.
With best regards,
Katja Breitkopf-Heinlein with all Co-authors.

Round 3
Reviewer 2 Report
The authors have made efforts to fix all the concerns that the manuscript quality has improved so much from the previous versions.